# Self-Supervised Spatio-Temporal Graph Learning for Point-of-Interest Recommendation

**Jiawei Liu** [1][ID]**, Haihan Gao** [2]**, Chuan Shi** [1,*]**, Hongtao Cheng** [1][ID] **and Qianlong Xie** [2]

[1] School of Computer Science (National Pilot Software Engineering School), Beijing University of Posts and Telecommunications, Beijing 100876, China; liu_jiawei@bupt.edu.cn (J.L.); 2019211200@bupt.cn (H.C.)

[2] Meituan, Beijing 100102, China; gaohaihan@meituan.com (H.G.); xieqianlong@meituan.com (Q.X.)

[*] Correspondence: shichuan@bupt.edu.cn

**Abstract:** As one of the most crucial topics in the recommendation system field, point-of-interest (POI) recommendation aims to recommending potential interesting POIs to users. Recently, graph neural networks have been successfully used to model interaction and spatio-temporal information in POI recommendations, but the data sparsity of POI recommendations affects the training of GNNs. Although some existing GNN-based POI recommendation approaches try to use social relationships or user attributes to alleviate the data sparsity problem, such auxiliary information is not always available for privacy reasons. Self-supervised learning provides a new idea to alleviate the data sparsity problem, but most existing self-supervised recommendation methods are designed for bi-partite graphs or social graphs, and cannot be directly used in the spatio-temporal graph of POI recommendations. In this paper, we propose a new method named SSTGL to combine self-supervised learning and GNN-based POI recommendation for the first time. SSTGL is empowered with spatio-temporal-aware strategies in the data augmentation and pre-text task stages, respectively, so that it can provide high-quality supervision information by incorporating spatio-temporal prior knowledge. By combining self-supervised learning objective with recommendation objectives, SSTGL can improve the performance of GNN-based POI recommendations. Extensive experiments on three POI recommendation datasets demonstrate the effectiveness of SSTGL, which performed better than existing mainstream methods.

**Keywords:** POI recommendation; graph neural network; self-supervised learning

## 1. Introduction

With the development of wireless communication and satellite positioning technologies, location-based services (LBSs) have become widely available in people's daily lives. LBSs can access the geographic coordinates of users through mobile devices and integrate them with other information (e.g., time and user preference) for services. Point-of-interest (POI) recommendation is a representative task in LBSs. Unlike typical recommendations that only consider the historical interaction between users and items, POI recommendations also need to consider the impact of spatio-temporal information [1] (e.g., the location coordinates of the POI, and the time of interaction between them).

Over the last decade, POI recommendation algorithms have evolved from early spatio-temporal-aware matrix factorization techniques [2–5] to recent approaches based on spatio-temporal graph representation learning [6–11]. As the most advanced graph representation learning technique, graph neural networks (GNNs), especially spatio-temporal graph neural networks (STGNNs), have been successfully applied to POI recommendations [9–11]. For example, as the first work applying a GNN to a POI recommendation, GPR [9] constructed additional POI–POI graphs based on the proximity of interaction times and uses a separate module to learn the representation of physical distances. As with other recommendation scenarios, POI recommendations also suffer from severe data sparsity [10].

Since GNN-based models usually rely on topology for message passing, the sparsity of interactions means that many nodes (especially long-tail nodes) cannot learn high-quality representations and are susceptible to interaction noise. To mitigate the impact of data sparsity on GNNs, some works have introduced various auxiliary information, including social relations [10] and user attributes [11]. However, as user privacy is taken more seriously these days, such auxiliary information is not always available. Therefore, there is an urgent need to explore new methods to reduce the impact of data sparsity problems on GNN-based approaches.

The recent emergence of self-supervised learning techniques offers a new direction to this problem. Through data augmentation and design pretext tasks, self-supervised learning can provide additional supervision signals to improve the performance of recommendation algorithms. In fact, some recent works have attempted to combine self-supervised learning with GNNs for other recommendation tasks (e.g., product recommendation [12,13], social recommendation [14], and session-based recommendation [15]). For example, SGL [12] designed three graph structure-based data augmentation methods, and NCL [13] designed graph structure-based and semantic prototype-based pretext tasks. However, these methods are specialized for bi-partite graphs, social graphs, or session-based hypergraphs, and do not consider spatio-temporal information. Thus, there is still a gap to adopt self-supervised learning for POI recommendation algorithms based on spatio-temporal graphs.

In this paper, we explore for the first time how self-supervised learning can be applied to GNN-based POI recommendations and design a general framework named self-supervised spatio-temporal graph learning (SSTGL). Specifically, SSTGL combines spatio-temporal prior knowledge in two ways, i.e., via data augmentation and the pre-text task. We first define the temporal and spatial similarity of POIs based on the interaction time between POIs and users as well as the geographical location of POIs. Then, considering that users may be interested in POIs that are similar to the interacted POIs, SSTGL adds some implicit edges to users and POIs based on the similarity function between POIs, which implements spatial or temporal-based data augmentation. Finally, SSTGL randomly drops some edges with a certain ratio to alleviate the data sparsity problem. For pre-text tasks, SSTGL uses spatio-temporal similarity to guide the consistency between node representations. Finally, we optimize the pre-text task, together with the recommendation ranking task, to improve the performance of the POI recommendation.

Experiments on three datasets show that our proposed model outperformed existing GNN-based POI recommendation algorithms. The relative improvements of Recall@50 were 6.32%, 13.27%, and 9.68%. In addition, ablation experiments and hyper-parameter experiments further demonstrated the robustness of our model.

The contributions of this paper are summarized as follows:

- To the best of our knowledge, this is the first attempt to design a self-supervised learning-based framework to improve GNN-based POI recommendation algorithms.
- We propose data augmentation strategies and pre-text tasks of the proposed framework, which model spatial or temporal prior knowledge from different perspectives.
- We conducted experiments on three POI recommendation datasets and verified that our model could improve GNN-based POI recommendations and outperform existing state-of-the-art methods.

## 2. Related Works

In this section, we review two related fields: point-of-interest (POI) recommendation and self-supervised learning (SSL).

### 2.1. Point-of-Interest Recommendation

Most of the traditional POI recommendation models are based on matrix factorization [2–5]. LRT [2] focused on the impact of temporal information on POI recommendation, while IRenMF [4] exploited geographic location information to model each location's neigh-

bors at both geographic location and geographic region levels. In addition, LGLMF [3] used local geographic information to obtain popular locations within the user's primary activity area. STACP [5] considered both geographic and temporal information, and they studied the behavior of users at different periods. Recently, some methods based on graph neural networks have been proposed [9–11]. For example, GPR [9] built the POI–POI graph by connecting adjacent POIs in the interaction sequence of users and POIs, and they used an exponential function to measure the physical distance between POIs to learn POI representations. HGMAP [10] additionally constructed a user–user graph based on social relationships, as well as a POI–POI graph based on geographic location, and combined the embedding of multiple graphs to obtain multiple user preference scores. GEAPR [11] learned user representations with the help of several factors, including user attributes, and used attention mechanisms to achieve interpretable recommendations. Data sparsity poses a great challenge to GNN-based recommendation [12]. Although social information and node attributes can alleviate the data sparsity problem to some extent, this auxiliary information is often not available due to the need to protect user privacy.

*2.2. Self-Supervised Learning*

Recently, as an effective way to alleviate the data sparsity problem, self-supervised learning (SSL) has been widely used in computer vision [16], natural language processing [17] and graph-based tasks, including various GNN-based recommendation tasks [12–15]. For product recommendation, SGL [12] applies SSL in recommendation tasks by changing the graph structure through a dropout and random walk strategy, as well as by maximizing the mutual information (MI) between multiple embeddings of the same node while minimizing the MI between the embeddings of different nodes. NCL [13] proposed structure-based and prototype-based contrastive learning objectives, which were used to improve graph collaborative filtering methods. For social recommendation, MHCN [14] designed a hypergraph convolutional network based on social relations and used hierarchical mutual information maximization to recover connectivity information in the hypergraph convolutional network. For session-based recommendation, S$^2$-DHCN [15] proposed a two-channel hypergraph convolutional network and maximized the mutual information between the learned session representations of both channels. However, these methods are not designed for POI recommendation, and they lack the use of spatio-temporal information, thus making them not applicable to spatio-temporal graphs.

## 3. Methodology

In this section, we first give the problem definition for the POI recommendation, and then introduce our proposed model SSTGL. As illustrated in Figure 1, our method consists of three main components, namely, the graph neural network backbone, the spatio-temporal-aware data augmentation, and the spatio-temporal-aware pre-text task.

In detail, the role of the graph neural network backbone is to learn the node representations from the user–POI graph $\mathcal{G}$ and use them for the recommendation loss $\mathcal{L}_{main}$ and the generation of the final recommendation list $\hat{P}_U$. Spatio-temporal aware data augmentation aims to generate multiple augmented graphs (i.e., $\mathcal{G}'$ and $\mathcal{G}''$) based on spatio-temporal prior knowledge, and these augmented graphs are also fed into the GNN backbone to generate multiple enhanced node representations. Spatio-temporal-aware pre-text tasks use spatio-temporal similarity to design optimization objectives $\mathcal{L}_{ssl}$ for self-supervised learning, thus providing effective self-supervised information.

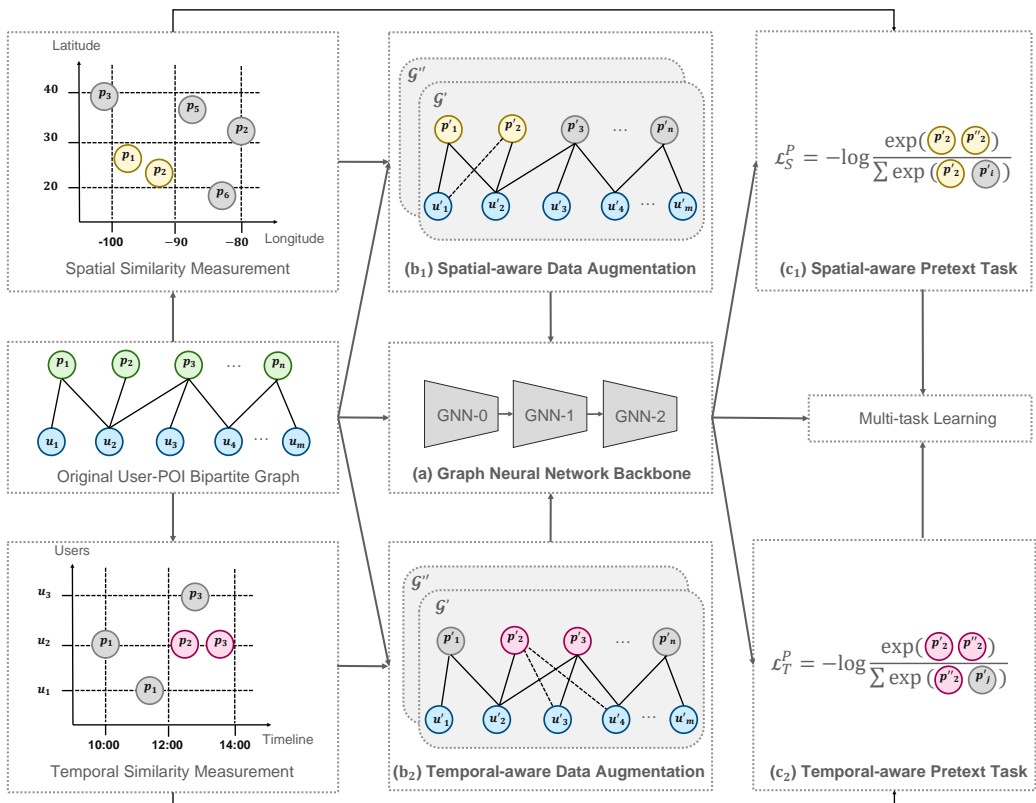

**Figure 1.** The proposed model SSTGL, which contains three main components. (**a**) Graph neural network backbone. (**b**) Spatial-temporal-aware data augmentation. (**c**) Spatial-temporal-aware pre-text task.

### 3.1. Problem Definition and GNN Backbone

In this paper, we modeled the POI recommendation as a link prediction problem on a bi-partite graph $\mathcal{G} = (\mathcal{V}, \mathcal{E})$, where users $\mathcal{U}$, POIs $\mathcal{P}$, nodes $\mathcal{V}$, and historical interactions **R** are between them as edges $\mathcal{E}$. Our goal was to find potential edges based on the observed edges and spatio-temporal information.

The GNN backbone module aims to generate the nodes' embeddings $\mathbf{Z}_U$ and $\mathbf{Z}_P$ by training a GNN function $f(\cdot | X_U, X_P)$ in a point-wise or pair-wise loss $\mathcal{L}_{main}$, where $X_U$ and $X_P$ are the nodes' features. Then, this module calculates the similarity score $\hat{Y}_{UP}$ between their embeddings and uses the top-K POIs with the highest similarity scores as the recommendation list $\hat{P}_U$.

In detail, graph neural networks usually use the message-passing mechanism to generate node representations. It consists of two specific steps. Firstly, given the bi-partite graph $\mathcal{G}$, the $(l+1)$-th layer representations of the nodes are updated by aggregating the $l$-th layer representations of their neighbor nodes:

$$\mathbf{Z}^{(l+1)} = f_{aggregate}(\cdot | \mathbf{Z}^{(l)}, \mathcal{G}), \tag{1}$$

where $\mathbf{Z}^{(0)} = X$ is a parameterized ID embedding lookup table.

Then, the representations of all the $L$ layers are fused to generate the final representations:

$$\mathbf{Z} = f_{readout}(\cdot | \mathbf{Z}^{(l)}, l = [0, \cdots, L]). \tag{2}$$

Although any $f_{aggregate}$ and $f_{readout}$ can be used as a GNN backbone, for the sake of fair comparison in subsequent experiments, SSTGL uses the same functions $f_{aggregate}(\cdot | z_1 \rightarrow z_2)$

and $f_{readout}(\cdot|z_1 \to z_2)$ as the self-supervised learning baselines [12,13] of the backbone, which can be defined as follows:

$$f_{aggregate}(\cdot|z_1 \to z_2) = \begin{cases} \sum_{p \in N_u} \frac{1}{\sqrt{|N_u||N_p|}} z_1 \, , z_1 = \mathbf{z}_p^{(l)} \text{and } z_2 = \mathbf{z}_u^{(l+1)}, \\ \sum_{u \in N_p} \frac{1}{\sqrt{|N_p||N_u|}} z_1 \, , z_1 = \mathbf{z}_u^{(l)} \text{and } z_2 = \mathbf{z}_p^{(l+1)}, \end{cases} \tag{3}$$

$$f_{readout}(\cdot|z_1 \to z_2) = \begin{cases} \frac{1}{L+1} \sum_{l=0}^{L} z_1 \, , z_1 = \mathbf{z}_u^{(l)} \text{and } z_2 = \mathbf{z}_u, \\ \frac{1}{L+1} \sum_{l=0}^{L} z_1 \, , z_1 = \mathbf{z}_p^{(l)} \text{and } z_2 = \mathbf{z}_p, \end{cases} \tag{4}$$

Finally, the similarity scores $\hat{Y}_{UP}$ of the representations of users and POIs are calculated using the inner product and are used to calculate the supervision loss $\mathcal{L}_{main}$:

$$\hat{Y}_{UP} = \mathbf{Z}_U^\top \mathbf{Z}_P, \tag{5}$$

$$\mathcal{L}_{main} = \sum_{(i,j,k) \in \mathcal{O}} -\log \sigma(\hat{y}_{ij} - \hat{y}_{ik}), \tag{6}$$

where $\sigma(\cdot)$ is the sigmoid function, $\mathcal{O} = \{(i,j,k)|(i,j) \in \mathcal{O}^+, (i,k) \in \mathcal{O}^-\}$ is the training data, $\mathcal{O}^+$ is positive pairs with interaction, and $\mathcal{O}^-$ is negative pairs without interaction.

### 3.2. Spatio-Temporal-Aware Data Augmentation

Some straightforward data augmentation operators have been proposed in the former work [12], including the node dropout, edge dropout, and random walk. However, these operators ignore the effect of spatio-temporal information, thus resulting in some low-quality graph structures, which in turn reduce the performance of POI recommendations. To take spatio-temporal information into account during data augmentation, in this section, we first defined the POI's temporal and spatial similarity matrices separately. Based on these similarity matrices, we defined the spatial-aware edge perturbation and temporal-aware edge perturbation operations. SSTGL uses these edge perturbation methods randomly at each epoch of training to generate new augmented graphs.

In detail, based on Equation (1), the aggregation operation based on the data augmentation can be expressed as follows:

$$\mathbf{Z}_1^{(l+1)} = f_{aggregate}(\cdot|\mathbf{Z}_1^{(l)}, s_1(\mathcal{G})), \\ \mathbf{Z}_2^{(l+1)} = f_{aggregate}(\cdot|\mathbf{Z}_2^{(l)}, s_2(\mathcal{G})), \tag{7}$$

where $s_1(\cdot)$ and $s_2(\cdot)$ are data augmentation operators. Note that, although the data augmentation can generate any number of views, SSTGL uses only two data augmentation operations to reduce the model complexity:

- Spatial similarity matrix $\mathcal{M}_S \in \{0,1\}^{|\mathcal{P}| \times |\mathcal{P}|}$: when the distance between two POIs is less than a certain threshold $K_S$, then the similarity of these two POIs is 1; otherwise, it is 0.
- Temporal similarity matrix $\mathcal{M}_T \in \{0,1\}^{|\mathcal{P}| \times |\mathcal{P}|}$: when two POIs have interacted with the same user in a period $K_T$, then the similarity of these two POIs is 1; otherwise, it is 0.

In practice, we set $K_T$ to 2 h in following the former work [18], which discovered that users tended to visit the same POI consecutively within 2 h. In addition, we used the Geohash algorithm to transform the geographic coordinates of the POIs into region IDs and then determined whether the IDs were the same as a threshold condition of $K_S$.

Based on the spatial and temporal similarity matrices, we defined the following data augmentation operators:

- Spatial-aware edge perturbation (SEP): It adds multiple implicit edges based on the spatial similarity to the original user–POI edges:

$$\mathcal{E}_{imp_S}^{|\mathcal{U}|\times|\mathcal{P}|} = \mathcal{E} \cdot \mathcal{M}_S, \tag{8}$$

$$s_1(\mathcal{G}) = (\mathcal{V}, \mathbf{\Theta_1}(\mathcal{E}_{imp_S}, \mathcal{E})), \ s_2(\mathcal{G}) = (\mathcal{V}, \mathbf{\Theta_2}(\mathcal{E}_{imp_S}, \mathcal{E})), \tag{9}$$

where $\mathbf{\Theta}_1(a, b)$, and $\mathbf{\Theta}_2(a, b)$ are perturbation vectors that pick elements from $a$ in some ratio $r_\theta$ and add it to $b$.

- Temporal-aware edge perturbation (TEP): It adds multiple implicit edges based on temporal similarity to the original user–POI edges:

$$\mathcal{E}_{imp_T}^{|\mathcal{U}|\times|\mathcal{P}|} = \mathcal{E} \cdot \mathcal{M}_T, \tag{10}$$

$$s_1(\mathcal{G}) = (\mathcal{V}, \mathbf{\Theta_1}(\mathcal{E}_{imp_T}, \mathcal{E})), \ s_2(\mathcal{G}) = (\mathcal{V}, \mathbf{\Theta_2}(\mathcal{E}_{imp_T}, \mathcal{E})). \tag{11}$$

In following [12], we used the above data augmentation approaches at each epoch to generate multiple views and used the same ratio $r_\theta$ on $\mathbf{\Theta}_1$ and $\mathbf{\Theta}_2$. We leave it as future work to use more than two data augmentation operators $s(\cdot)$ simultaneously and to use different perturbation ratios $r_\theta$ for different operators.

### 3.3. Spatio-Temporal-Aware Pre-Text Task

For contrastive self-supervised learning, the pre-text tasks $\mathcal{L}_{ssl}$ are defined based on positive and negative pairs. Existing methods usually treat representations of the same node with different augmentation methods [12] or with different GNN layers [13] as positive pairs, as well as different nodes using different augmentation methods [12] or different GNN layers [13] as negative pairs; neither of these approaches takes into account the spatio-temporal information. In this section, we propose spatial-aware and temporal-aware pre-text tasks, which are further used to define spatial-aware and temporal-aware contrastive learning objectives.

In detail, we believe the spatio-temporal prior knowledge gives some clues to selecting positive and negative samples. For example, POIs with similar spatio-temporal properties are more suitable for positive samples than negative samples. Due to data sparsity, these similar POIs may not have interacted with the same user and could easily be mistaken as negative pairs, thus reducing the recommendation performance. Therefore, we designed spatial-aware and temporal-aware pre-text tasks, which are defined as follows:

- Spatial-aware pre-text task (SPT): We took $Q_S$ nodes with the highest spatial similarity in $\mathcal{M}_S$ to the target node as positive examples and $\rho(\%)$ nodes with the lowest spatial similarity as negative examples.
- Temporal-aware pre-text task (TPT): We took $Q_T$ nodes with the highest temporal similarity in $\mathcal{M}_T$ to the target node as positive examples and $\rho(\%)$ nodes with the lowest temporal similarity as negative examples.

Based on the above definitions, we proposed the spatial-aware and temporal-aware contrastive learning objectives:

- Spatial-aware contrastive learning (SCL): Maximizes the MI between spatial-aware positive POI pairs and minimizes the MI between spatial-aware negative POI pairs:

$$\mathcal{L}_S^P = \sum_{i\in\mathcal{P}} -\log\frac{\sum_{j\in Q_S} \exp\left(s\left(\mathbf{z}_i^1, \mathbf{z}_j^2\right)/\tau\right)}{\sum_{k\in(|\mathcal{P}|\cdot\rho_S)} \exp\left(s\left(\mathbf{z}_i^1, \mathbf{z}_k^2\right)/\tau\right)}, \tag{12}$$

where $s(\cdot)$ is a similarity measure function, which is set as a dot product; $\tau$ is the temperature in softmax; and $(\cdot)^1$ and $(\cdot)^2$ represent different embeddings obtained from data augmentation.

- Temporal-aware contrastive learning (TCL): Maximizes the MI between temporal-aware positive POI pairs and minimizes the MI between temporal-aware negative POI pairs.

$$\mathcal{L}_T^P = \sum_{i \in \mathcal{P}} -\log \frac{\sum_{j \in Q_T} \exp\left(s\left(\mathbf{z}_i^1, \mathbf{z}_j^2\right)/\tau\right)}{\sum_{k \in (|\mathcal{P}| \cdot \rho_T)} \exp\left(s(\mathbf{z}_i^1, \mathbf{z}_k^2)/\tau\right)}. \tag{13}$$

In practice, we set $Q_S$ and $Q_T$ to 1; that is, SSTGL only used the target node as a positive example. We leave larger scales or different $Q_S$ and $Q_T$ values to future work. In addition, since SSTGL did not define a user-based spatio-temporal similarity matrix, it used a spatio-temporal independent contrastive learning objective for users as in [12]:

$$\mathcal{L}^U = \sum_{u \in \mathcal{U}} -\log \frac{\exp\left(s\left(\mathbf{z}_u^1, \mathbf{z}_u^2\right)/\tau\right)}{\sum_{v \in \mathcal{U}} \exp\left(s(\mathbf{z}_u^1, \mathbf{z}_v^2)/\tau\right)}. \tag{14}$$

Using POI-based and user-based self-supervised objective functions, we defined the final self-supervised optimization objective as follows:

$$\mathcal{L}_{ssl} = \mathcal{L}^U + \alpha \mathcal{L}^P, \tag{15}$$

where $\alpha$ is a hyper-parameter used to balance two losses, and $\mathcal{L}^P \in \{\mathcal{L}_S^P, \mathcal{L}_T^P\}$.

### 3.4. Model Training

To use self-supervised signals to improve the performance of the POI recommendations, SSTGL uses a multi-task training strategy to optimize the ranking loss and contrastive learning loss jointly:

$$\mathcal{L} = \mathcal{L}_{main} + \lambda_1 \mathcal{L}_{ssl} + \lambda_2 \|\Phi\|_2^2, \tag{16}$$

where $\lambda_1$ and $\lambda_2$ are hyper-parameters used to control the strengths of the SSL and regularization term, and $\Phi$ denotes the set of GNN parameters.

### 3.5. Complexity Analyses of SSTGL

We show the algorithm in Algorithm 1. Since SSTGL introduces no trainable parameters, its and space complexity remain the same as the GNN backbone. In addition, without any change to the neural network structure, its inference time complexity also remains the same as the GNN backbone. Therefore, the main extra time complexity comes from computing the self-supervised loss. Let $|E|$ and $|V|$ be the number of edges and nodes in the user–item graph, respectively, $s$ denote the number of epochs, $B$ denote the batch size, and $d$ denote the embedding size. The time complexity of the whole training phase is $O(|E|d(2 + |V|s))$, which is the same as an existing self-supervised GNN model [12].

---

**Algorithm 1** The framework of SSTGL

---

**Require:** Given the original user-POI graph $\mathcal{G} = (\mathcal{V}, \mathcal{E})$, the layer $L$ of GNN model.
**Ensure:** The similarity scores $\hat{Y}_{UP}$.
 1: Initialize all node embeddings $\mathbf{Z}^{(0)}$ and compute similarity scores by Equation (7);
 2: **while** not converge **do**
 3: 　**for** each layer $l$ in $[0, \ldots, L-1]$ **do**
 4: 　　Aggregate neighbor information to update node embeddings $\mathbf{Z}^{(l+1)}$ by Equation (3);
 5: 　**end for**
 6: 　Compute final node embeddings $\mathbf{Z}$ using readout strategy by Equation (4);
 7: 　Compute similarity scores $\hat{Y}_{UP}$ by Equation (5);
 8: 　Compute recommendation loss $\mathcal{L}_{main}$ by Equation (6);
 9: 　Compute self-supervised loss $\mathcal{L}_{ssl}$ by Equation (15);
 10: 　Backpropagate based on overall loss $\mathcal{L}$ computed by Equation (16).
 11: **end while**

---

## 4. Experiments

To verify the validity of SSTGL and explore the reasons behind it, in this section, we conducted extensive experiments to answer the following research questions (RQs):

- RQ1: How does the proposed SSTGL method perform when compared with the start-of-the-art baselines?
- RQ2: How does each component of the SSTGL contribute to the overall performance?
- RQ3: How do different hyper-parameters influence the performance of SSTGL?

### 4.1. Experimental Setup

4.1.1. Datasets

We conduct experiments on three benchmark datasets: Foursquare, Gowalla, and Meituan. The statistics of the datasets are shown in Table 1.

- Foursquare [19]: The Foursquare dataset consists of check-in data generated on Foursquare from April 2012 to September 2013. Following [19], we removed users with less than 10 interactions and POIs with less than 10 interactions. After preprocessing, it contained 1,196,248 check-ins between 24,941 users and 28,593 POIs.
- Gowalla [19]: The Gowalla dataset consists of check-in data generated on Gowalla from February 2009 to October 2010. As was done in [19], we removed users with less than 15 interactions and POIs with less than 10 interactions. After preprocessing, it contained 1,278,274 check-ins between 18,737 users and 32,510 POIs.
- Meituan "https://www.biendata.xyz/competition/smp2021_1/ (accessed on 28 July 2023)": The Meituan dataset consists of check-in data generated on the Meituan APP from 1st March 2021 to 28th March 2021. We removed users with less than 10 interactions and POIs without location information. After preprocessing, it contained 602,331 check-ins between 38,904 users and 3182 POIs.

**Table 1.** Dataset statistics.

| Dataset | #Check-Ins | #POI | #User | Sparsity | Time Span |
|---------|-----------|------|-------|----------|-----------|
| Foursquare | 1,196,248 | 28,593 | 24,941 | 99.90% | April 2012–September 2013 |
| Gowalla | 1,278,274 | 32,510 | 18,737 | 99.87% | February 2009–October 2010 |
| Meituan | 602,331 | 3182 | 38,904 | 99.51% | 1st March 2021–28th March 2021 |

For each dataset, we chose the oldest 70% of interactions of each user as the training data and the newest 20% of interactions as the test data. The remaining 10% were used as validation data.

4.1.2. Baselines

We compared SSTGL with the following ten models, which can be classified according to Table 2:

- NeuMF [20]: NeuMF is a classical MF-based model that combines matrix factorization and multi-layer perceptron to learn both low-dimensional and high-dimensional embeddings.
- NGCF [21]: NGCF is a GNN-based model capturing high-order information through message passing and aggregation.
- DGCF [22]: DGCF is a GNN-based model, which models different relationships and separates user intents in the representation.
- LightGCN [23]: LightGCN is a GNN-based recommendation model, which simplifies the aggregation step by deleting the weight matrix and activation function.
- SGL [12]: SGL is a graph-based self-supervised method that proposes three data augmentation strategies based on the graph structure.
- NCL [13]: NCL is a graph-based contrastive learning method that improves neural graph collaborative filtering by considering structural and semantic neighbors.

- LGLMF [3]: LGLMF is an MF-based POI recommendation model, which combines logistic matrix factorization with a region-based geographical model.
- STACP [5]: STACP is also an MF-based POI recommendation model, which combines matrix factorization with a spatio-temporal activity-centers algorithm.
- GPR [9]: GPR is a GNN-based model designed for POI recommendation that uses an extra POI–POI graph to learn item embeddings and improve performance.
- MPGRec [24]: MPGRec is the newest GNN-based POI recommendation model, which uses a dynamic memory module to store global information for spatial consistency.

**Table 2.** Category of baselines and SSTGL.

| | **MF-Based** | **GNN-Based** | **GNN and SSL-Based** |
|---|---|---|---|
| ST-unaware | NeuMF [20] | NGCF [21], DGCF [22], LightGCN [23] | SGL [12], NCL [13] |
| ST-aware | LGLMF [3], STACP [5] | GPR [9], MPGRec [24] | SSTGL(Ours) |

For all these baselines, we followed the default hyper-parameter settings as stated in their papers. Note that, since our datasets do not have auxiliary information such as social relationships or user attributes, we did not use the methods from [10,11,25,26] that utilize the aforementioned auxiliary information as baselines. In addition, since we focused on the classical POI recommendation task, we did not choose GNN models for the next POI recommendation [27,28] or tour the recommendations [29] as baselines.

### 4.1.3. Evaluation Metrics

To evaluate the model's performance, we adopted two general metrics, Recall@N and mean average precision (MAP@N), where N means the top-N POIs recommended by the model. In our experiments, N was set to 5, 10, 20, and 50 for a comprehensive comparison.

### 4.1.4. Implementation Details

We implemented SSTGL based on Recbole [30], which is a PyTorch-based open-source framework to develop recommendation algorithms. As for the model training, we employed the Adam [31] optimizer to minimize the overall loss $\mathcal{L}$, where we set the learning rate to $lr = 0.005$ for the Foursquare and Gowalla datasets and to $lr = 0.05$ for the Meituan dataset. For SSTGL and all the baselines, we adopted an early-stopping strategy with a patience of 10 epochs to prevent overfitting, where Recall@5 was the indicator. We set the balance hyper-parameters to $\alpha = 1$, $\lambda_1 = 0.05$, and $\lambda_2 = 0.00005$. We tuned the hyper-parameters to $\tau \in [0.1, 0.2, 0.5, 1]$, $r_\theta \in [0.1, 0.3, 0.5, 0.7, 0.9]$, and $\rho \in [0.01, 0.05, 0.1, 0.2]$.

### 4.2. Performance Comparison (RQ1)

The experimental results on the three datasets are shown in Table 3, Table 4 and Table 5, respectively. Note that there were four different variants of SSTGL depending on the data augmentation and the choice of the pre-text task:

- SSTGL(SEP): uses spatial-aware edge perturbation and non-spatio-temporal pre-text tasks.
- SSTGL(TEP): uses temporal-aware edge perturbation and non-spatio-temporal pre-text task.
- SSTGL(SCL): uses spatial-aware contrastive learning and non-spatio-temporal data augmentation.
- SSTGL(TCL): use temporal-aware contrastive learning and non-spatio-temporal data augmentation.

**Table 3.** Overall performance of SSTGL and all baselines for Foursquare dataset, where the best results for each column are shown in bold font, and the second-place results are underlined.

| Model | Recall@5 | Recall@10 | Recall@20 | Recall@50 | MAP@5 | MAP@10 | MAP@20 | MAP@50 |
|---|---|---|---|---|---|---|---|---|
| NeuMF | 0.0368 | 0.0610 | 0.0981 | 0.1727 | 0.0235 | 0.0241 | 0.0271 | 0.0310 |
| NGCF | 0.0390 | 0.0627 | 0.0980 | 0.1688 | 0.0249 | 0.0250 | 0.0278 | 0.0314 |
| DGCF | 0.0435 | 0.0669 | 0.1028 | 0.1764 | 0.0291 | 0.0288 | 0.0316 | 0.0354 |
| LightGCN | 0.0469 | 0.0721 | 0.1076 | 0.1796 | 0.0317 | 0.0312 | 0.0341 | 0.0378 |
| SGL | 0.0452 | 0.0707 | 0.1080 | 0.1852 | 0.0300 | 0.0299 | 0.0330 | 0.0371 |
| NCL | 0.0463 | 0.0723 | 0.1083 | 0.1839 | 0.0313 | 0.0310 | 0.0338 | 0.0378 |
| STACP | 0.0274 | 0.0450 | 0.0700 | 0.1275 | 0.0187 | 0.0186 | 0.0206 | 0.0235 |
| LGLMF | 0.0284 | 0.0459 | 0.0729 | 0.1284 | 0.0192 | 0.0190 | 0.0212 | 0.0242 |
| GPR | 0.0316 | 0.0502 | 0.0763 | 0.1272 | 0.0183 | 0.0205 | 0.0224 | 0.0243 |
| MPGRec | **0.0592** | 0.0848 | 0.1200 | 0.1915 | **0.0366** | **0.0398** | **0.0425** | **0.0452** |
| SSTGL (Ours) | 0.0577 | **0.0851** | **0.1244** | **0.2036** | 0.0338 | 0.0374 | 0.0401 | 0.0427 |

**Table 4.** Overall performance of SSTGL and all baselines for Gowalla dataset, where the best results for each column are shown in bold font, and the second-place results are underlined.

| Model | Recall@5 | Recall@10 | Recall@20 | Recall@50 | Map@5 | Map@10 | Map@20 | Map@50 |
|---|---|---|---|---|---|---|---|---|
| NeuMF | 0.0302 | 0.0497 | 0.0808 | 0.1478 | 0.0227 | 0.0211 | 0.0231 | 0.0267 |
| NGCF | 0.0308 | 0.0500 | 0.0810 | 0.1458 | 0.0235 | 0.0216 | 0.0234 | 0.0268 |
| DGCF | 0.0332 | 0.0530 | 0.0834 | 0.1477 | 0.0266 | 0.0239 | 0.0254 | 0.0288 |
| LightGCN | 0.0352 | 0.0564 | 0.0897 | 0.1593 | 0.0271 | 0.0247 | 0.0267 | 0.0305 |
| SGL | 0.0338 | 0.0557 | 0.0911 | 0.1657 | 0.0259 | 0.0240 | 0.0262 | 0.0305 |
| NCL | 0.0344 | 0.0561 | 0.0902 | 0.1631 | 0.0267 | 0.0244 | 0.0264 | 0.0305 |
| STACP | 0.0176 | 0.0302 | 0.0509 | 0.0964 | 0.0142 | 0.0131 | 0.0143 | 0.0168 |
| LGLMF | 0.0241 | 0.0398 | 0.0646 | 0.1156 | 0.0209 | 0.0188 | 0.0201 | 0.0230 |
| GPR | 0.0302 | 0.0483 | 0.0766 | 0.1310 | 0.0183 | 0.0196 | 0.0216 | 0.0238 |
| MPGRec | 0.0471 | 0.0706 | 0.1050 | 0.1718 | **0.0308** | 0.0325 | 0.0349 | 0.0377 |
| SSTGL (Ours) | **0.0511** | **0.0786** | **0.1175** | **0.1946** | 0.0292 | **0.0329** | **0.0357** | **0.0383** |

In detail, we used dropedge [12] as the non-spatial-temporal data augmentation, and we defined the non-spatio-temporal pre-text task of the POI in a similar way to Equation (14). We present the best results among these variants with respect to the performance of SSTGL in the tables. The differences in performance between the variants are analyzed in the next section.

Based on the experimental results, we have the following observations:

- SSTGL outperformed all baseline methods in most cases. In particular, the relative improvements from the strongest baselines were 6.32% (Foursquare), 13.27% (Gowalla), and 9.68% (Meituan) using the Recall@50 metric. Note that SSTGL not only worked better than the existing POI recommendation methods, but also better than the existing self-supervised graph learning methods. This demonstrates the ability of our model to use self-supervised learning to alleviate the data sparsity problem in the POI recommendation task. Although MPGRec performed better in some cases, it relies on a dynamic memory module, which requires a large memory overhead.

- For the baseline models, the GNN models did not always outperform the MF models, which was related to the datasets and model architectures. For example, we found that the NeuMF model performed better than some GNN-based methods for the Meituan dataset. This may be due to the low sparsity of the Meituan dataset and the more personalized interests of the users in the take-out scenario, so aggregating higher-order neighborhood information would instead reduce the performance.

**Table 5.** Overall performance of SSTGL and all baselines for Meituan dataset, where the best results for each column are shown in bold font, and the second-place results are underlined.

| Model | Recall@5 | Recall@10 | Recall@20 | Recall@50 | MAP@5 | MAP@10 | MAP@20 | MAP@50 |
|---|---|---|---|---|---|---|---|---|
| NeuMF | 0.3540 | 0.4315 | 0.4631 | 0.5063 | 0.2339 | 0.2517 | 0.2562 | 0.2588 |
| NGCF | 0.3198 | 0.3952 | 0.4499 | 0.5177 | 0.2123 | 0.228 | 0.2349 | 0.2392 |
| DGCF | 0.3285 | 0.4046 | 0.4612 | 0.5335 | 0.2194 | 0.2353 | 0.2427 | 0.2472 |
| LightGCN | 0.3456 | 0.4221 | 0.4705 | 0.5315 | 0.2250 | 0.2417 | 0.2482 | 0.252 |
| SGL | 0.3373 | 0.4123 | 0.4674 | 0.5311 | 0.2432 | 0.2591 | 0.2664 | 0.2705 |
| NCL | 0.3466 | 0.4236 | 0.4667 | 0.5187 | 0.2222 | 0.2391 | 0.2451 | 0.2484 |
| STACP | 0.0054 | 0.0094 | 0.0203 | 0.0421 | 0.0022 | 0.0026 | 0.0034 | 0.0041 |
| LGLMF | 0.0005 | 0.0010 | 0.0030 | 0.0087 | 0.0002 | 0.0002 | 0.0004 | 0.0005 |
| GPR | 0.2984 | 0.3758 | 0.4573 | 0.5669 | 0.2066 | 0.2185 | 0.2253 | 0.2297 |
| MPGRec | **0.3930** | <u>0.4316</u> | <u>0.4655</u> | <u>0.5147</u> | **0.3170** | **0.3240** | **0.3272** | **0.3294** |
| SSTGL (Ours) | <u>0.3865</u> | **0.4388** | **0.4917** | **0.5645** | <u>0.3073</u> | <u>0.3146</u> | <u>0.3185</u> | <u>0.3209</u> |

### 4.3. Ablation Study (RQ2)

To explore the effects of different data augmentation and pre-text tasks on the model performance, we show the results of four model variants in Figure 2. Based on the results, we have the following observations:

- In the strategies we designed, the temporal-based approaches (i.e., TEP and TCL) worked better on the Foursquare and Gowalla datasets, and the spatial-aware approaches (i.e., SEP and SCL) performed better on the Meituan dataset. This may indicate that the spatial factor has a greater influence on the Meituan dataset compared with other datasets.
- Although both consider spatio-temporal information, the data augmentation-based approach outperformed the pre-text-task-based approach. This may be due to the direct modification of the graph structure using the self-supervised method of data augmentation, which allows the node representation of the GNN output to make better use of spatio-temporal prior knowledge.

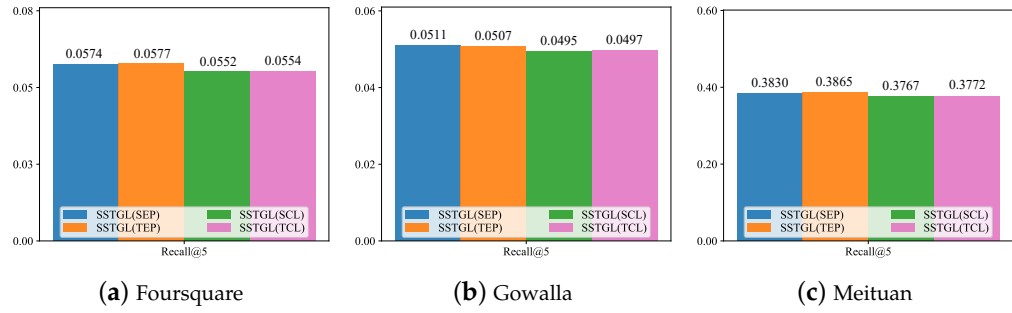

**(a)** Foursquare     **(b)** Gowalla     **(c)** Meituan

**Figure 2.** Ablation study on Foursquare, Gowalla, and Meituan datasets.

### 4.4. Influence of Hyper-Parameters (RQ3)

Our model contained three main hyper-parameters (i.e., the drop ratio $r_\theta$, the sample ratio $\rho$, and the temperature $\tau$). In Figure 3, we show the performance of the model for the Foursquare dataset with different values of hyper-parameters. We have the following observations:

- Overall, the different drop ratio $r_\theta$ and sample ratio $\rho$ had little effect on the model results, which indicates the robustness of the model.
- Too large or too small SSL temperature $\tau$ values reduced the performance. This observation is consistent with the previous work [12]. The possible reason behind this is that, if the temperature is large, it is more difficult to distinguish negative examples. If the temperature is small, only a small number of negative cases affect the optimization.

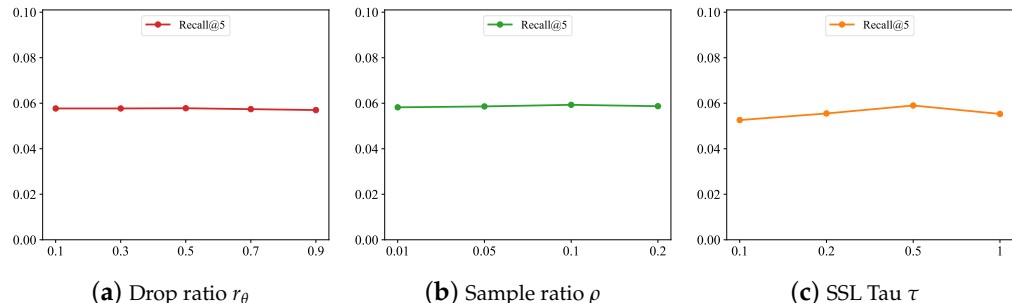

**Figure 3.** Hyper-parameter experiments for Foursquare dataset.

*4.5. Discussion*

Based on our experiments, we found that existing self-supervised learning strategies did not always improve the GNN models. For example, the SGL performed worse than the LightGCN in most cases. This indicates that data augmentation approaches that do not take into account the spatio-temporal information may not help POI recommendations to alleviate the data sparsity problem. In contrast, our model took spatio-temporal information into account when designing the self-supervised method, so the performance was improved.

In addition, we were surprised to find that the spatio-temporal approaches did not always outperform the non-spatio-temporal models. In particular, the STACP and LGLMF performed extremely poorly on the Meituan dataset, which may be due to the fact that these two methods rely on the geographic coordinates of the POI, while the Meituan dataset only has the region ID of the POI. Since POI recommendation algorithms are often not compared with non-spatio-temporal recommendation algorithms, many works ignore that recommendation algorithms based on bi-partite graphs can also be very strong baselines. In practical application, this also inspires us to fuse the spatio-temporal information for the POI recommendation based on the existing non-spatio-temporal GNNs (e.g., LightGCN), which may be more effective and efficient than designing completely new spatio-temporal GNNs (e.g., GPR).

**5. Conclusions**

In this paper, we proposed a novel self-supervised spatio-temporal graph learning model (SSTGL) to improve the GNN's potential for use in POI recommendations. In particular, we designed a model-agnostic self-supervised learning framework that took into account spatio-temporal prior knowledge. Based on the framework, we defined spatio-temporal aware data augmentation and pre-text tasks. Extensive experiments showed that SSTGL outperformed existing methods for the POI recommendation task.

In future work, we will extend more views and more flexible data augmentation strategies under the self-supervised learning framework, and we will try to apply graph-based self-supervised learning to more complex tasks, including next POI recommendation and tour recommendation.

**Author Contributions:** Conceptualization, J.L. and H.G.; methodology, J.L. and C.S.; software, H.G. and Q.X.; validation, J.L. and H.C.; formal analysis, J.L.; investigation, J.L.; resources, C.S.; data curation, H.G.; writing—original draft preparation, J.L.; writing—review and editing, H.G. and C.S.; visualization, H.C.; supervision, C.S.; project administration, Q.X.; funding acquisition, C.S. All authors have read and agreed to the published version of the manuscript.

**Funding:** This research was funded by the National Natural Science Foundation of China (No. U20B2045, U1936220, 62192784, 62172052, 62002029, 61772082).

**Institutional Review Board Statement:** Not applicable.

**Informed Consent Statement:** Not applicable.

**Data Availability Statement:** The data presented in this study are openly available in [19] and https://www.biendata.xyz/competition/smp2021_1/.

**Conflicts of Interest:** The authors declare no conflict of interest.

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
