# Peer review of "Self-Supervised Spatio-Temporal Graph Learning for Point-of-Interest Recommendation"

_applsci, doi:10.3390/app13158885_

Round 1

Reviewer 1 Report

[1] In section 3.2 (Spatio-temporal aware Data Augmentation), Authors set Kt 2 hrs to transform the geographic coordinates of POIs into region IDs. Why? Proper justification is citation is required.

[2] Line no. 192, In practice? Give proper citation.

[3] Author may refer following two latest works for POI recommendations: [i] Acharya M, et al. How can we create a recommender system for tourism? A location centric spatial binning-based methodology using social networks. International Journal of Information Management Data Insights. 2023 Apr 1;3(1):100161. [ii] Differential Privacy-Based Social Network Detection Over Spatio-Temporal Proximity for Secure POI Recommendation. SN Computer Science. 2023 Mar 7;4(3):252.

[3] It is suggested to include algorithms / pseudo code of the proposed approach.

[4] The authors should provide a complexity analysis of the proposed method.

[5] Overall discussion is required in in the context of POI recommendation. A discussion section may be added before conclusion.

[6] The writing needs to be improved.

NA

Author Response

Point 1: In section 3.2 (Spatio-temporal aware Data Augmentation), Authors set Kt 2 hrs to transform the geographic coordinates of POIs into region IDs. Why? Proper justification is citation is required.

Response 1: Thank you for the insightful comment. We have decided to adopt your suggestions, and have revised the paper in the corresponding places.

[Paragraph 3, Section 3.2] “In practice, we set $K_T$ to 2 hours following the former work[1], which discovers users tend to visit the same POI consecutively within 2 hours.”

[1] Han, H.; Zhang, M.; Hou, M.; Zhang, F.; Wang, Z.; Chen, E.; Wang, H.; Ma, J.; Liu, Q. STGCN: a spatial-temporal aware graph learning method for POI recommendation. In Proceedings of the 2020 IEEE International Conference on Data Mining (ICDM), 2020.

Point 2: Line no. 192, In practice? Give proper citation.

Response 2: Thank you for the insightful comment. We have decided to adopt your suggestions, and have revised the paper in the corresponding places.

[Paragraph 5, Section 3.2] In practice, following [2], we use the above data augmentation approaches at each epoch to generate multiple views and use the same ratio.

[2] Wu, J.; Wang, X.; Feng, F.; He, X.; Chen, L.; Lian, J.; Xie, X. Self-supervised graph learning for recommendation. In Proceedings of the Proceedings of the 44th international ACM SIGIR conference on research and development in information retrieval, 2021, pp. 726–735.

Point 3: Author may refer following two latest works for POI recommendations: [i] Acharya M, et al. How can we create a recommender system for tourism? A location centric spatial binning-based methodology using social networks. International Journal of Information Management Data Insights. 2023 Apr 1;3(1):100161. [ii] Differential Privacy-Based Social Network Detection Over Spatio-Temporal Proximity for Secure POI Recommendation. SN Computer Science. 2023 Mar 7;4(3):252.

Response 3: Thank you for the insightful comment. We have decided to adopt your suggestions, and have revised the paper in the corresponding places.

[Paragraph 2, Section 4.1.2] “Note that since our datasets does not have auxiliary information such as social relationships or user attributes, we do not use methods [3,4,5,6] that utilize the aforementioned auxiliary information as baselines.”

[5] Acharya M, et al. How can we create a recommender system for tourism? A location centric spatial binning-based methodology using social networks. International Journal of Information Management Data Insights. 2023 Apr 1;3(1):100161.

[6] Differential Privacy-Based Social Network Detection Over Spatio-Temporal Proximity for Secure POI Recommendation. SN Computer Science. 2023 Mar 7;4(3):252.

Point 4: It is suggested to include algorithms / pseudo code of the proposed approach.

Response 4: Thank you for the insightful comment. We have decided to adopt your suggestions, and have added the algorithm in the Section 3.5.

Point 5: The authors should provide a complexity analysis of the proposed method.

Response 5: Thank you for the insightful comment. We have decided to adopt your suggestions, and have added Section 3.5 to analyze complexity.

Point 6: Overall discussion is required in in the context of POI recommendation. A discussion section may be added before conclusion.

Response 6: Thank you for the great and insightful comment. We have decided to adopt your suggestions, and have added Section 4.5 to give an overall discussion.

Point 7: The writing needs to be improved.

Response 7: Thank you for the insightful comment. We have decided to adopt your suggestions, and have embellished the writing by adding several new sections (i.e. Section 3.5 and Section 4.5) ,reformatting the Table 1 and Table 2 and revising multiple grammar errors.

Reviewer 2 Report

In the content, including the abstract, I would advise using the impersonal form to emphasize the objective nature of the scientific study.

In this paper, authors combine self-supervised learning and GNN-based POI recommendation for the first time. They design spatio-temporal aware strategies in the data augmentation and pretext task stages, respectively, so that it can provide high-quality supervision information incorporating spatio-temporal prior knowledge.

The authors conducted a literature review and showed related works. They proposed an interesting solution and presented it clearly. They posed three research questions. At the end of the article, the discussion and conclusion should be developed. The theoretical and practical implications of the topic discussed should be clearly presented.

Author Response

Point 1: In the content, including the abstract, I would advise using the impersonal form to emphasize the objective nature of the scientific study.

Response 1: Thank you for the insightful comment. We have decided to adopt your suggestions, and have revised the paper in several places by using “this approach” or “SSTGL” rather than personal form.

Point 2: At the end of the article, the discussion and conclusion should be developed. The theoretical and practical implications of the topic discussed should be clearly presented.

Response 2: Thank you for the insightful comment. We have decided to adopt your suggestions, and have added Section 4.5 to give an overall discussion.

Round 2

Reviewer 1 Report

My concerns have been addressed